# On Misinformation Containment in
# Online Social Networks

**Guangmo (Amo) Tong**
Department of Computer and Information Sciences
University of Delaware
amotong@udel.edu

**Weili Wu**
Department of Computer Science
University of Texas at Dallas
weiliwu@utdallas.edu

**Ding-Zhu Du**
Department of Computer Science
University of Texas at Dallas
dzdu@utdallas.edu

## Abstract

The widespread online misinformation could cause public panic and serious economic damages. The misinformation containment problem aims at limiting the spread of misinformation in online social networks by launching competing campaigns. Motivated by realistic scenarios, we present an analysis of the misinformation containment problem for the case when an arbitrary number of cascades are allowed. This paper makes four contributions. First, we provide a formal model for multi-cascade diffusion and introduce an important concept called as cascade priority. Second, we show that the misinformation containment problem cannot be approximated within a factor of $\Omega(2^{\log^{1-\epsilon} n^4})$ in polynomial time unless $NP \subseteq DTIME(n^{\text{polylog} \, n})$. Third, we introduce several types of cascade priority that are frequently seen in real social networks. Finally, we design novel algorithms for solving the misinformation containment problem. The effectiveness of the proposed algorithm is supported by encouraging experimental results.

## 1 Introduction

The past years have witnessed a drastic increase in the usage of online social networks. By the end of April 2018, there are totally 3.03 billion active social media users and each Internet user has an average of 7.6 social media accounts [24]. Despite allowing efficient exchange of information, online social networks have provided platforms for misinformation. Misinformation may lead to serious economic consequences and even cause panics. For example, it was reported by NDTV that the misinformation on social media led to Pune violence in January 2018.[1] Recently, the rapid spread of misinformation has been on the list of top global risks according to World Economic Forum [2]. Therefore, effective strategies on misinformation control are imperative.

Information propagates through social networks via cascades and each cascade starts to spread from certain seed users. When misinformation is detected, a feasible strategy is to launch counter campaigns competing against the misinformation [1]. Such counter campaigns are usually called as positive cascades. The misinformation containment (MC) problem aims at selecting seed users for positive cascades such that the misinformation can be effectively restrained. The existing works

have considered this problem for the case when there is one misinformation cascade and one positive cascade [2, 3, 4]. In this paper, we address this problem for the general case when there are multiple misinformation cascades and positive cascades. The scenario considered in this paper is more realistic because there always exists multiple cascades concerning one issue or news in a real social network.

**Example 1.** In the 2016 US presidential election, the fake news that Hillary Clinton sold weapons to ISIS has been widely shared in online social networks. More than 20 articles spreading this fake news were discovered on Facebook in October 2016 [5]. While these articles all supported the fake news, they were spreading on Facebook as different information cascades because they had different sources and exhibited different levels of reliability. On the other hand, multiple articles aiming at correcting this fake news were being shared by the users standing for Hillary Clinton. These articles can be taken as the positive cascades and, again, they spread as individual cascades. The model proposed in this paper applies to such a scenario.

We introduce an important concept, called as *cascade priority*, which defines how the users make selections when more than one cascades arrive at the same time. As shown later, the cascade priority is a necessary and critical setting when multiple cascades exist. The model proposed in this paper is a natural extension of the existing models, but the MC problem becomes very challenging under the new setting. For example, as shown later in Sec. 5, adding more seed nodes for the positive cascade may surprisingly cause a wider spread of misinformation, i.e., the objective function is not monotone nondecreasing. Our goal in this paper is to offer a systematic study, including formal model formulation, hardness analysis, and algorithm design. The contributions of this paper are summarized as follows.

- We provide a formal model supporting multi-cascade influence diffusion in online social networks. To the best of our knowledge, we are the first to consider the issue on cascade priority. Based on the proposed model, we study the MC problem by formulating it as a combinatorial optimization problem.
- We prove that the MC problem under the general model cannot be approximated within a factor of $\Omega(2^{\log^{1-\epsilon} n^4})$ in polynomial time unless $NP \subseteq DTIME(n^{\mathrm{polylog}\, n})$.[3]
- We propose and study three types of cascade priorities, *homogeneous cascade priority*, *M-dominant cascade priority*, and *P-dominant cascade priority*, as defined in Sec. 5. These special cascade priorities are commonly seen in real social networks, and the MC problem enjoys desirable combinatorial properties under these settings.
- We design a novel algorithm for the MC problem by using nontrivial upper bound and lower bound. As shown in the experiments, the proposed algorithm outperforms other methods and it admits a near-constant data-dependent approximation ratio on all the considered datasets.

## 2   Related work.

**Influence maximization (IM).** The influence maximization (IM) problem is proposed by Kempe, Kleinberg, and Tardos in [6] where the authors also develop two basic diffusion models, independent cascade (IC) model and linear threshold (LT) model. It is shown in [6] that the IM problem is actually a submodular maximization problem and therefore the greedy scheme provides a $(1 - 1/e)$-approximation. However, Chen *et al.* in [7] prove that it is #P-hard to compute the influence and the naive greedy algorithm is not scalable to large datasets. One breakthrough is made by C. Borgs *et al.* [8] who invent the reverse sampling technique and design an efficient algorithm. This technique is later improved by Tang *et al.* [9] and Nguyen *et al.* [10]. Recently, Li *et al.* [18] study the IM problem under non-submodular threshold functions and Lynn *et al.* [19] consider the IM problem under the Ising network. For the continuous-time generative model, N. Du *et al.* [30] propose a scalable influence estimation method and then study the IM problem under the continuous setting.

**Misinformation containment (MC).** Based on the IC and LT model or their variants, the MC problem is then proposed and extensively studied. Budak *et al.* [2] consider the independent cascade model and show that the MC problem is again a submodular maximization problem when there are two cascades. Tong *et al.* [4, 31] design an efficient algorithm by utilizing the reverse sampling

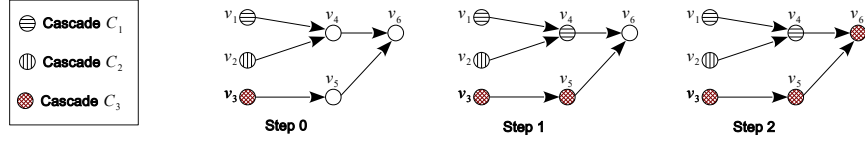

Figure 1: An illustrative example of diffusion process.

technique. He *et al.* [3], Fan *et al.* [11] and Zhang *et al.* [12] study the MC problem under competitive linear threshold model. Nguyen *et al.* [13] propose the IT-Node Protector problem which limits the spread of misinformation by blocking the high influential nodes. Different from the existing works, we focus on the general case when more than two cascades are allowed. In other contexts, He *et al.* [20] study the MC problem in mobile social networks and Wang *et al.* [21] study the MC problem with the consideration of user experience. Mehrdad *et al.* [28] consider a point process network activity model and study the fake news mitigation problem by reinforcement learning. Recently, a comprehensive survey [29] regarding false information is provided by Srijan *et al.*

## 3 Model and problem formulation

In this section, we formally formulate the diffusion model and the MC problem.

### 3.1 Model

A social network is given by a directed graph $G = (V, E)$. For each edge $(u, v)$, we say $v$ is an out-neighbor of $u$, and $u$ is an in-neighbor of $v$. Information is assumed to spread via cascades and each cascade spreads from seed users. Let $\mathbb{C}$ be the set of all the cascades, and we use $\tau(C) \subseteq V$ to denote the seed set of a cascade $C \in \mathbb{C}$. We say a user is $C$-active if they are activated by cascade $C$. All users are initially defined as $\emptyset$-active. Associated with each edge $(u, v)$, there is a real number $p_{(u,v)} \in [0, 1]$ denoting the propagation probability from $u$ to $v$. We assume that $p_{(u,v)} = 0$ iff $(u, v) \notin E$. When $u$ becomes $C$-active for a certain cascade $C \in \mathbb{C}$, they attempt once to activate an $\emptyset$-active out-neighbor $v$ with the success probability of $p_{(u,v)}$. We assume that a user will be activated by the cascade arriving first and will not be activated later for another time. Associated with each user $v$, each cascade $C$ is given a unique priority, denoted by $\mathrm{F}_v(C)$, which gives a linear order over the cascades. $\mathrm{F}_v$ can be represented as a bijection between $\mathbb{C}$ and $\{1, 2, ..., |\mathbb{C}|\}$, and, for each $C_1, C_2 \in C$, $\mathrm{F}_v(C_1) > \mathrm{F}_v(C_2)$ iff $C_1$ has a higher priority than that of $C_2$ at $v$. If two or more cascades reach $v$ at the same time, $v$ will be activated by the cascade with the highest priority. The cascade priority at each node is affected by many factors such as the reputation of the source, the reliability of the message and the user's personal opinion. Several special cascade priorities will be introduced later in Sec. 5.

For a time step $t \in \{0, 1, 2, ...\}$, we use $\pi_t(v) \in \mathbb{C} \cup \{\emptyset\}$ to denote the activation state of a user $v$ after time step $t$, where $\pi_t(v) = C$ (resp. $\pi_t(v) = \emptyset$) if $v$ is $C$-active (resp. $\emptyset$-active). Let $\pi_\infty(v)$ be the activation state of $v$ when the diffusion process terminates. The diffusion process unfolds stochastically in discrete, described as follows:

- Time step 0. If a node $v$ is selected as a seed node by one or more cascades, $v$ becomes $C^*$-active where $C^* = \arg\max_{C \in \{C | C \in \mathbb{C}, v \in \tau(C)\}} \mathrm{F}_v(C)$ .
- Time step t. Each node $u$ activated at time step $t - 1$ attempts to activate each of $u$'s $\emptyset$-active out-neighbor $v$ with a success probability of $p_{(u,v)}$. If a node $v$ is successfully activated by one or more in-neighbors, $v$ becomes $\pi_{t-1}(u^*)$-active where $u^* = \arg\max_{u \in A \subseteq V} \mathrm{F}_v(\pi_{t-1}(u))$ where $A$ is the set of the in-neighbors who successfully activate $v$ at time step $t$.[4]

**Example 2.** Consider the network shown in Fig. 1 where there are three cascades $C_1$, $C_2$ and $C_3$, of which the seed sets are $\{v_1\}$, $\{v_2\}$ and $\{v_3\}$, respectively. Suppose that $p_e = 1$ for each edge $e$, $\mathrm{F}_{v_4}(C_1) > \mathrm{F}_{v_4}(C_2)$, and, $\mathrm{F}_{v_6}(C_2) > \mathrm{F}_{v_6}(C_3) > \mathrm{F}_{v_6}(C_1)$. At time step 1, $v_4$ becomes $C_1$-active due to that $\mathrm{F}_{v_4}(C_1) > \mathrm{F}_{v_4}(C_2)$. Because $\mathrm{F}_{v_6}(C_3) > \mathrm{F}_{v_6}(C_1)$, $v_6$ is finally $C_3$-active. One can see that $v_6$ would be $C_2$-active if the cascade priority at $v_4$ was $\mathrm{F}_{v_4}(C_2) > \mathrm{F}_{v_4}(C_1)$.

## 3.2 Problem formulation

We assume that, regarding one issue or topic, there are two groups of cascades: misinformation cascades and positive cascades. Suppose there are already some cascades in the network and their seed sets are known to us. For the purpose of misinformation containment, we launch a new positive cascade with a certain seed set. We use $\mathbb{M}$ and $\mathbb{P}$ to denote the sets of the existing misinformation cascades and positive cascades, respectively, and use $P_*$ to denote the newly introduced positive cascade. Therefore, $\mathbb{C} = \mathbb{M} \cup \mathbb{P} \cup \{P_*\}$. We say a user is $\mathbb{M}$-active if they are $M$-active for some $M \in \mathbb{M}$, otherwise they are called as $\overline{\mathbb{M}}$-active.[5] For a seed set $\tau(P_*)$ of cascade $P_*$, we use $f_{\mathbb{M}}(\tau(P_*))$ (resp. $f_{\overline{\mathbb{M}}}(\tau(P_*))$) to denote the expected number of the $\mathbb{M}$-active (resp. $\overline{\mathbb{M}}$-active) nodes when the diffusion process terminates. The problems considered in this paper are shown as follows.

**Problem 1** (Min-$\mathbb{M}$ problem). Given a budget $k \in \mathbb{Z}^+$ and a candidate set $V^* \subseteq V$, select a seed set $\tau(P_*) \subseteq V^*$ for $P_*$ with $|\tau(P_*)| \leq k$ such that $f_{\mathbb{M}}(\tau(P_*))$ is minimized.

Alternatively, we can maximize the number of the $\overline{\mathbb{M}}$-active users.

**Problem 2** (Max-$\overline{\mathbb{M}}$ problem). Given a budget $k \in \mathbb{Z}^+$ and a candidate $V^* \subseteq V$, select a seed set $\tau(P_*) \subseteq V^*$ for $P_*$ with $|\tau(P_*)| \leq k$ such that $f_{\overline{\mathbb{M}}}(\tau(P_*))$ is maximized.

An instance of the above problems is given by (1) $G = (V, E)$: a network structure; (2) $\{p_e | p_e \in [0, 1], e \in E\}$: the probabilities on the edges; (3) $\mathbb{C} = \mathbb{M} \cup \mathbb{P} \cup \{P_*\}$: the set of the existing cascades together with $P_*$; (4) $\{F_v(C) | v \in V, C \in \mathbb{C}\}$: the cascade priority at each node; (5) $\{\tau(C) \subseteq V | C \in \mathbb{M} \cup \mathbb{P}\}$: the seed sets of the existing cascades; (6) $V^* \subseteq V$: a candidate set of the seed nodes of $P_*$. The propagation probability and the cascade priority can be inferred by mining historical data [25, 26, 27].

**Remark 1.** When $|\mathbb{M}| = 1$, it becomes the model considered in [14]. When $|\mathbb{M}| = 1$, $|\mathbb{P}| = 0$ and the cascade priority is homogeneous[6], the problem considered in [2, 4] reduces to the Max-$\overline{\mathbb{M}}$ problem.

## 4 Hardness result

In this section, we provide a hardness result for the Min-$\mathbb{M}$ problem. The result is obtained by a reduction from the positive-negative partial set cover ($\pm$PSC) problem.

**Problem 3** ($\pm$PSC problem). An instance of $\pm$PSC is a triplet $(X, Y, \Phi)$ where $X$ and $Y$ are two sets of elements with $X \cap Y = \emptyset$, and $\Phi = \{\phi_1, ..., \phi_m\} \subseteq 2^{X \cup Y}$ is collection of subsets over $X \cup Y$. For each $\Phi^* \subseteq \Phi$, its cost is defined as $|X \setminus (\cup_{\phi \in \Phi^*} \phi)| + |Y \cap (\cup_{\phi \in \Phi^*} \phi)|$. The $\pm$PSC problem seeks for a $\Phi^* \subseteq \Phi$ with the minimum cost.

The following result is presented by Miettinen [15].

**Lemma 1** ([15]). *There exists no polynomial-time approximation algorithm for $\pm$PSC with an approximation factor of $\Omega(2^{\log^{1-\epsilon} |m|^4})$ for any $\epsilon > 0$, unless $NP \subseteq DTIME(n^{\mathrm{polylog}\, n})$.*

A core result is given in the next lemma.

**Lemma 2.** *For any $\alpha(|V^*|) > 1$, $\pm$PSC is approximable to within a factor of $4 \cdot \alpha(m) - 3$, if Min-$\mathbb{M}$ is approximable to within a factor of $\alpha(|V^*|)$.*

*Proof.* For an arbitrary instance $(X, Y, \Phi)$ of the $\pm$PSC problem, we construct an instance of the Min-$\mathbb{M}$ problem accordingly, as shown in Fig. 2.

**The graph.** Let us first construct the graph $G$. For each $x_i \in X$, we add a node $x_i$ to the graph, and for each $y_i \in Y$ we add two nodes $y_i$ and $z_i$ to the graph. For each $\phi_i \in \Phi$, we add a node $\phi_i$ to the graph. We further add four nodes $a$, $b_1$, $b_2$ and $c$, as shown in Fig. 2. For each $\phi_i$ and $x_j$ (resp. $y_j$), we add an edge $(\phi_i, x_j)$ (resp. $(\phi_i, y_j)$) iff $x_j \in \phi_i$ (resp. $y_j \in \phi_i$). For each $z_i$, we add an edge $(y_i, z_i)$ and an edge $(c, z_i)$. We add an edge $(a, y_i)$ for each $y_i \in Y$ and an edge $(b_2, x_i)$ for each $x_i \in X$. Finally, we add an edge $(b_1, c)$. The probability of each edge is set as 1.

---

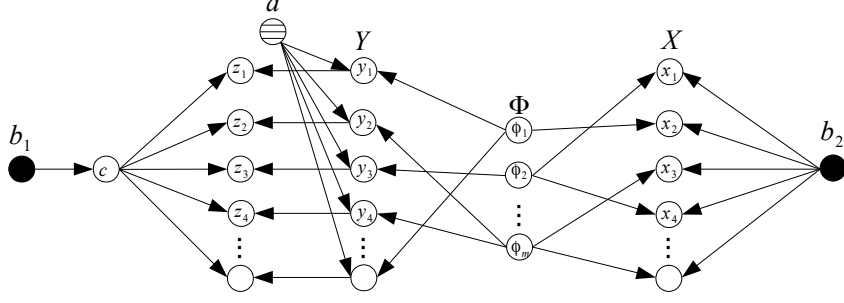

[5]Note that an $\emptyset$-active node is $\overline{\mathbb{M}}$-active.

[6]The definition of homogeneous cascade priority is given later in Sec. 5.

Figure 2: Reduction.

**Cascade setting.** We assume there is one misinformation cascade $M_1$ with the seed set $\{b_1, b_2\}$ and one positive cascade $P_1$ with the seed set $\{a\}$. We aim at introducing one positive cascade $P_*$ by selecting at most $k = m$ seed nodes from $V^* = \Phi = \{\phi_1, ...., \phi_m\}$. For each $y_i \in \{y_1, ..., y_m\}$, the cascade priority is set as $F_{y_i}(P_*) > F_{y_i}(P_1)$. For each node $z_i$ in $\{z_1, ..., z_m\}$, the cascade priority is set as $F_{z_i}(P_1) > F_{z_i}(M_1) > F_{z_i}(P_*)$. The cascade priority at other nodes can be set arbitrarily.

**Analysis.** Each set $\Phi^* \subseteq \Phi$ corresponds to a solution to the Min-$\mathbb{M}$ problem. We use $g(\Phi^*)$ to denote the objective function of the $\pm$PSC problem, i.e.,

$$g(\Phi^*) = |X \setminus (\cup_{\phi \in \Phi^*} \phi)| + |Y \cap (\cup_{\phi \in \Phi^*} \phi)|.$$

Now let us fix $\Phi^*$ and analyze the activation state of the nodes. Note that each node $y_i$ will be either $P_1$-active or $P_*$-active. In particular, $y_i$ is $P_*$ active iff $y_i$ is in some $\phi \in \Phi^*$. Furthermore, according the cascade priority at $z_i$, $z_i$ is $\mathbb{M}$-active iff $y_i$ is $P_1$-active. Therefore, $z_i$ is $\overline{\mathbb{M}}$-active iff $y_i$ is not in $\cup_{\phi \in \Phi^*} \phi$. For each node $x_i \in X$, it is $\overline{\mathbb{M}}$-active iff it is in some $\phi \in \Phi^*$. Finally, it can be easily checked that the nodes in $\Phi \cup Y \cup \{a\}$ will be $\overline{\mathbb{M}}$-active and the nodes in $\{c, b_1, b_2\}$ will be $\mathbb{M}$-active, regardless of $\Phi^*$. As a result,

$$f_{\mathbb{M}}(\Phi^*) = 3 + |X \setminus \cup_{\phi \in \Phi^*} \phi| + |Y \cap \cup_{\phi \in \Phi^*} \phi| = 3 + g(\Phi^*).$$

Thus, $OPT \subseteq \Phi$ is an optimal solution to the MC instance iff $OPT$ is an optimal solution to the instance of the $\pm$PSC problem. Suppose that $\Phi^*$ is an $\alpha(|V^*|)$-approximation to the Min-$\mathbb{M}$ problem for some $\alpha(|V^*|) > 1$. We have

$$f_{\mathbb{M}}(\Phi^*) \leq \alpha(|V^*|) \cdot f_{\mathbb{M}}(OPT) \iff 3 + g(\Phi^*) \leq \alpha(|V^*|) \cdot (3 + g(OPT))$$

$$\iff \frac{g(\Phi^*)}{g(OPT)} \leq \alpha(|V^*|) + \frac{3(\alpha(|V^*|) - 1)}{g(OPT)} \implies \frac{g(\Phi^*)}{g(OPT)} \leq 4\alpha(|V^*|) - 3.$$

Since $|V^*| = |\Phi| = m$, $\Phi^*$ is a $(4 \cdot \alpha(m) - 3)$-approximation to the instance of the $\pm$PSC problem. $\square$

The following result follows immediately from Lemmas 1 and 2.

**Theorem 1.** *For any $\epsilon > 0$, there is no polynomial-time approximation algorithm for the Min-$\mathbb{M}$ problem with an approximation factor of $\Omega(2^{\log^{1-\epsilon} |V^*|^4})$ unless $NP \subseteq DTIME(n^{\text{polylog } n})$.*

## 5 Algorithms

In this section, we present algorithms for the Max-$\overline{\mathbb{M}}$ problem. Throughout this section, we denote the objective function $f_{\overline{\mathbb{M}}}$ as $f$. The technique of submodular maximization has been extensively used in the existing works. For a set function $h()$ over a ground set $U$, the properties of monotone nondecreasing and submodular are defined as follows:

**Definition 1** (Monotone nondecreasing). $h(A) \leq h(B)$, for each $A \subseteq B \subseteq U$.

**Definition 2** (Submodular). $h(A) + h(B) \geq h(A \cup B) + h(A \cap B)$, for each $A, B \subseteq U$.

As mentioned in Remark 1, the Max-$\overline{\mathbb{M}}$ problem is a natural extension of the problem considered in [2, 4], but it is not submodular and even not monotone nondecreasing.

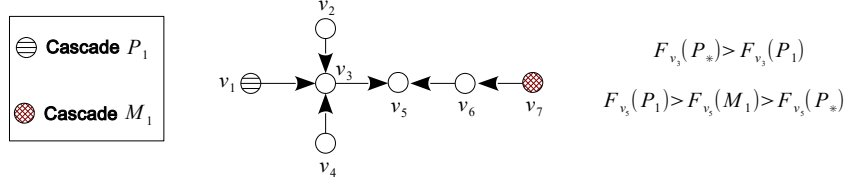

Figure 3: An illustrative example of non-submodularity.

**Example 3.** Consider the network shown in Fig. 3, where there exists one positive cascade $P_1$ and one misinformation cascade $M_1$. Now we deploy a new positive cascade $P_*$ and assume the candidate seed set $V^*$ is equal to $V$. Suppose that the probability on each edge is equal to 1, $\tau(P_1) = \{v_1\}$ and $\tau(M_1) = \{v_7\}$, and the cascade priority at $v_3$ and $v_5$ is given as shown in the figure. We can observe that $f(\{\emptyset\}) = 5$, $f(\{v_2\}) = f(\{v_4\}) = f(\{v_2, v_4\}) = 4$. Therefore, $f(\{v_2\}) < f(\emptyset)$, and $f(\{v_2\}) + f(\{v_4\}) < f(\{v_2\} \cap \{v_4\}) + f(\{v_2\} \cup \{v_4\})$. This illustrates that inappropriately selecting positive seed nodes may lead to a wider spread of misinformation.

In the rest of this section, we first study three special cascade priorities and then design an algorithm for the general setting.

### 5.1 Special cases: homogeneous, M-dominant and P-dominant cascade priority

We introduce the following types of cascade priority that frequently appear in real social networks.

**Definition 3** (Homogeneous cascade priority). The cascade priority is said to be homogeneous if $F_v(C) = F_u(C)$ for each $u, v \in V$ and $C \in \mathbb{C}$. That is, each cascade has the same priority at each node.

**Definition 4** (M-dominant cascade priority). The cascade priority is said to be M-dominant if $F_v(M) > F_v(P)$ for each $M \in \mathbb{M}$, $P \in \mathbb{P} \cup \{P_*\}$ and $v \in V$. Informally speaking, at each node, the priority of each misinformation cascade is higher than that of any positive cascade.

Similarly, we have the P-dominant cascade priority.

**Definition 5** (P-dominant cascade priority). The cascade priority is said to be P-dominant if $F_v(P) > F_v(M)$ for each $M \in \mathbb{M}$, $P \in \mathbb{P} \cup \{P_*\}$ and $v \in V$.

**Remark 2.** The homogeneous cascade priority is capable of representing the case when the priority of cascade is determined by the source or the initiator of the cascade. For example, when there are two opposite cascades $C_1$ and $C_2$ regarding NBA on Twitter, where $C_1$ is posted by ESPN while $C_2$ comes from an unknown source, the users will all tend to believe $C_1$ and therefore $F_v(C_1) > F_v(C_2)$ for each $v \in V$. The M-dominant or P-dominant cascade priority describes the scenario when one group of the cascades are well polished and very convincing. For example, the fake news in Example 1 was believed to be true by many online users because it was claimed to be released by WikiLeaks. As a result, the fake news always had a higher cascade priority and $F_v(M) > F_v(P)$ for each $M \in \mathbb{M}$, $P \in \mathbb{P} \cup \{P_*\}$ and $v \in V$.

While the Max-$\overline{\mathbb{M}}$ problem does not exhibit any good property in general, it is indeed monotone nondecreasing and submodular under special cascade priority settings. For the above types of cascade priority, we have the following results.

**Theorem 2.** *$f$ is monotone nondecreasing and submodular if the cascade priority is M-dominant or P-dominant.*

**Theorem 3.** *$f$ is monotone nondecreasing and submodular if the cascade priority is homogeneous.*

Please see the supplementary material for the proofs of Theorems 2 and 3. Note that the greedy algorithm yields a $(1 - 1/e)$-approximation when the objective function is monotone nondecreasing and submodular [16]. Theorems 2 and 3 evince that special cascade priorities may admit desirable combinatorial properties. In the next subsection, we will utilize these results to design an effective algorithm for the Max-$\overline{\mathbb{M}}$ problem for the general case.

---

**Algorithm 1** Greedy scheme

---

1: **Input:** a function $h$ over a ground set $U$ and a budget $k$;
2: $U_0 \leftarrow \emptyset$;
3: **for** $i = 1 : k$ **do**
4:      $u \leftarrow \arg\max_{u \in U} h(U_{i-1} \cup \{u\}) - h(U_{i-1})$;
5:      $U_i \leftarrow U_{i-1} \cup \{u\}$;
6: **return** $U \leftarrow \arg\max_{U_i} h(U_i)$;

---

**Algorithm 2** Sandwich approximation strategy

---

1: **Input:** $f, \overline{f}, \underline{f}, V^*, k$;
2: $\overline{S}_* \leftarrow$ ALG. $1(\overline{f}, V^*, k)$; $\underline{S}_* \leftarrow$ ALG. $1(\underline{f}, V^*, k)$; $S_* \leftarrow$ ALG. $1(f, V^*, k)$;
3: **return** $S' = \arg\max_{S \in \{\overline{S}_*, \underline{S}_*, S_*\}} f(S)$;

---

## 5.2 General case

For the general cascade priority, we present a data-dependent approximation algorithm based on the upper-lower-bound technique [22]. Each cascade priority $F_v$ induces another two cascade priorities, defined as follows:

**Definition 6** ($\overline{F}_v$). $\overline{F}_v$ is a cascade priority at node $v$ induced by $F_v$, satisfying,

(a) for each $P_1, P_2 \in \mathbb{P} \cup \{P_*\}$, $\overline{F}_v(P_1) < \overline{F}_v(P_2) \iff F_v(P_1) < F_v(P_2)$,

(b) for each $M_1, M_2 \in \mathbb{M}$, $\overline{F}_v(M_1) < \overline{F}_v(M_2) \iff F_v(M_1) < F_v(M_2)$, and,

(c) for each $P \in \mathbb{P} \cup \{P_*\}$ and $M \in \mathbb{M}$, $\overline{F}_v(M) < \overline{F}_v(P)$.

**Definition 7** ($\underline{F}_v$). $\underline{F}_v$ is a cascade priority at node $v$ induced by $F_v()$, satisfying (a) and (b) in Def. 6, and, for each $P \in \mathbb{P} \cup \{P_*\}$ and $M \in \mathbb{M}$, $\underline{F}_v(P) < \underline{F}_v(M)$.

$\overline{F}_v$ and $\underline{F}_v$ keep the relative priority of the cascades within the same group and adjust the relative priority of the cascades between groups. We can easily check that $\overline{F}_v$ and $\underline{F}_v$ are uniquely determined by $F_v$.

**Example 4.** Suppose there are three positive cascades, $P_1, P_2$ and $P_3$, and two misinformation cascades, $M_1$ and $M_2$. If $F_v(P_3) < F_v(P_1) < F_v(M_2) < F_v(P_2) < F_v(M_1)$, then we have $\overline{F}_v(M_2) < \overline{F}_v(M_1) < \overline{F}_v(P_3) < \overline{F}_v(P_1) < \overline{F}_v(P_2)$ and $\underline{F}_v(P_3) < \underline{F}_v(P_1) < \underline{F}_v(P_2) < \underline{F}_v(M_2) < \underline{F}_v(M_1)$.

For a seed set $\tau(P_*) \subseteq V^*$ of cascade $P_*$, we use $\overline{f}(\tau(P_*))$ (resp. $\underline{f}(\tau(P_*))$) to denote the expected number of the $\overline{\mathbb{M}}$-active nodes when each node $v$ replaces its cascade priority $F_v$ by $\overline{F}_v$ (resp. $\underline{F}_v$). Because $\overline{F}_v$ is P-dominant and $\underline{F}_v$ is M-dominant, the following result immediately follows from Theorem 2.

**Corollary 1.** *$\overline{f}$ and $\underline{f}$ are both monotone nondecreasing and submodular.*

Furthermore, $\overline{f}$ is an upper bound of $f$ and $\underline{f}$ is a lower bound of $f$.

**Theorem 4.** *For each $\tau(P_*) \subseteq V^*$, $\overline{f}(\tau(P_*)) \geq f(\tau(P_*)) \geq \underline{f}(\tau(P_*))$.*

Please see the supplementary material for the proof of Theorem 4. We now present an algorithm to solve the Max-$\overline{\mathbb{M}}$ problem by approximating $\overline{f}$ and $\underline{f}$. First, we run the greedy algorithm, ALG. 1, on all three functions, $f, \overline{f}$ and $\underline{f}$, to obtain three solutions $S_*, \overline{S}_*$ and $\underline{S}_*$, respectively. The final solution is selected as $S' = \arg\max_{S \in \{S_*, \overline{S}_*, \underline{S}_*\}} f(S)$. The process is formally shown in ALG. 2. According to [22], it has the following performance bound.

**Theorem 5.** *$f(S') \geq \max\{\frac{f(\overline{S}_*)}{\overline{f}(\overline{S}_*)}, \frac{\underline{f}(OPT)}{f(OPT)}\} \cdot (1 - 1/e) \cdot f(OPT)$, where $OPT$ is the optimal solution.*

**Remark 3.** The performance bound of ALG. 2 depends on the closeness of the upper bound and the lower bound. We will experimentally examine this gap in Sec. 6.

# 6 Experiments

In this section, we evaluate the proposed algorithm by experiments. Our goal is to examine the performance of ALG. 2 by (a) comparing it to baseline methods and (b) measuring the data-dependent approximation ratio given in Theorem 5. Our experiments are performed on a server with a 2.2 GHz eight-core processor.

## 6.1 Setup

**Dataset.** The first dataset, collected from Twitter, is built after monitoring the spreading process of the messages posted between 1st and 7th July 2012 regarding the discovery of a new particle with the features of the elusive Higgs boson [17]. It consists of a collection of activities between users, including re-tweeting action, replying action, and mentioning action. We extract two subgraphs from this dataset, where the first one has 10,000 nodes and the second one has 100,000 nodes. We denote these two graphs by Higgs-10K and Higgs-100K, respectively. The second dataset, denoted by HepPh, is a citation graph from the e-print arXiv with 34,546 papers [23]. HepPh has been widely used in the study on influence diffusion in social networks. The statistics of the datasets can be found in the supplementary material.

**Propagation Probability.** On Higss-10K, the probability of edge $(u, v)$ is set to be proportional to the frequency of the activities between $u$ and $v$. In particular, we set $p_{(u,v)}$ as $\frac{a_i}{a_{max}} \cdot p_{max} + p_{base}$, where $a_i$ is the number of activities from $u$ to $v$, $a_{max}$ is the maximum number of the activities among all the edges, and, $p_{max} = 0.2$ and $p_{base} = 0.4$ are two constants. On Higgs-100K, we adopt the uniform setting where the propagation probability on each edge is set as 0.1. On HepPh, we adopt the wighted cascade setting and set $p_{(u,v)}$ as $1/deg(v)$ where $deg(v)$ is the number of in-neighbors of $v$. The uniform setting and the weighted cascade are two classic settings and they have been widely used in the existing works [2, 4, 6, 7, 9, 10, 18].

**Cascade setting.** We consider three cases where there are three cascades, five cascades and ten cascades, respectively. For the case of three cascades, we deploy one existing misinformation cascade and one existing positive cascade, and we launch a new positive cascade $P_*$. For each existing cascade, the size of the seed set is set as 20 and the seed nodes are selected from the node with the highest single-node influence. The seed sets of different cascades do not overlap with each other. The budget of $P_*$ is enumerated from $\{1, 2, ..., 20\}$ and the candidate set $V^*$ is equal to $V$. The cascade priority at each node is assigned randomly by generating a random permutation over $\{1, 2, 3\}$. We process the cases with five and ten cascades in the same way as the three cascades case. The details can be found in the supplementary material.

**Baseline methods.** Since there is no algorithm explicitly addressing the model considered in this paper, we consider three baseline methods, **HighWeight**, **Proximity** and **Random**. The weight of a node $v$ is defined as the sum of the probabilities of its out-edges (i.e., $\sum_v p_{(u,v)}$). HighWeight outputs the seed set according to the decreasing order of the node weight. Proximity selects the seed nodes of $P_*$ from the out-neighbors of the seed nodes of the misinformation cascades, where the preference is given to the node with a large weight. Random is a baseline method which selects the seed nodes randomly. The performance of Random is evaluated by the mean over 1,000 executions.

**Estimating influence.** The feasibility of ALG. 2 relies on the assumption that there is an efficient oracle of $f_{\overline{\mathbb{M}}}$. Unfortunately, it has been shown in [7] that computing the influence is a #P-hard problem, and in fact, it is also hard to compute $f_{\overline{\mathbb{M}}}$. In our experiments, the function value is estimated by 5,000 Monte Carlo simulations whenever $f_{\overline{\mathbb{M}}}$ is called, and the final solution of each algorithm is evaluated by 10,000 simulations. We note that the techniques proposed in [4, 8, 9, 10] are potentially applicable to the MC problem, but improving the efficiency of the algorithm is beyond the scope of this paper.

## 6.2 Result and discussion

The experimental results are shown in Figs. 4, 5 and 6. In each figure, the first three subfigures show the performance under the settings of three, five and ten cascades, respectively. Each subfigure gives four curves plotting the number of $\mathbb{M}$-active nodes under Sandwich (ALG. 2), HighWeight, Proximity and Random, respectively. The last subfigure shows the value of $f(\overline{S}_*)/\overline{f}(\overline{S}_*)$ in each experiment.

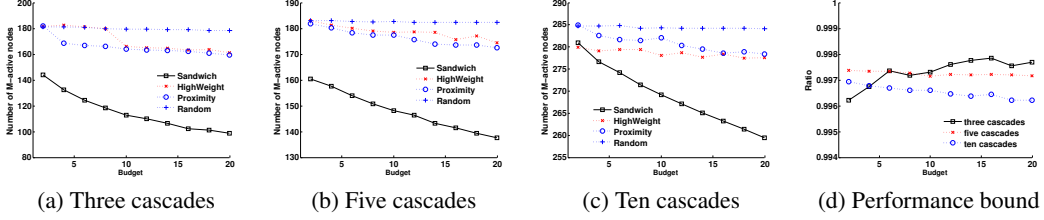

(a) Three cascades     (b) Five cascades     (c) Ten cascades     (d) Performance bound

Figure 4: Results on Higgs-10K.

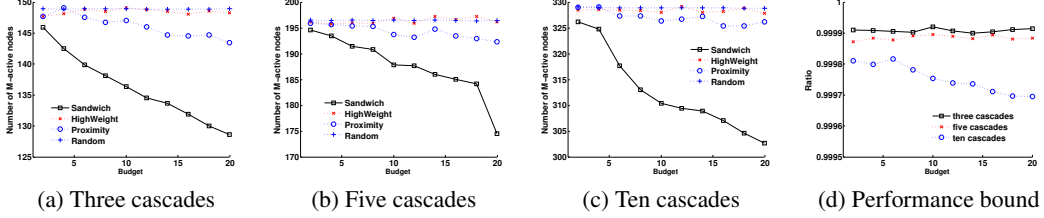

(a) Three cascades     (b) Five cascades     (c) Ten cascades     (d) Performance bound

Figure 5: Results on Higgs-100K.

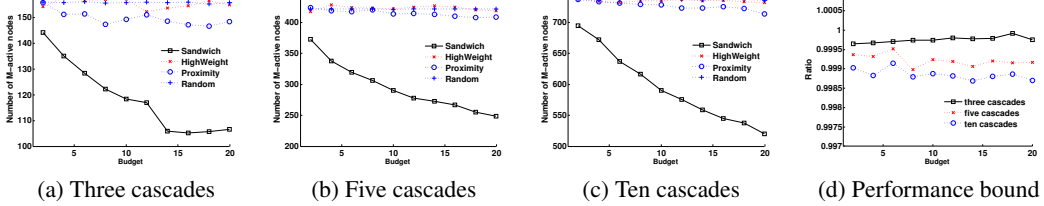

(a) Three cascades     (b) Five cascades     (c) Ten cascades     (d) Performance bound

Figure 6: Results on HepPh.

**Major observations.** First, as shown in the figures, ALG. 2 consistently provides the best performance. Comparing it to other baseline methods, the superiority of ALG. 2 can be very significant when the budget becomes large. As shown in Fig. 4a, on Higgs-10K, when there are three cascades and the budget is equal to 20, ALG. 2 is able to reduce the number of $\mathbb{M}$-active nodes from 180 to 100, while other methods can hardly make it below 160. Another important observation is that the ratio $f(\overline{S}_*)/\overline{f}(\overline{S}_*)$ is very close to 1 in practice. For example, on HepPh, this ratio is always larger than 0.9985. This means the performance ratio of ALG. 2 is guaranteed to be very close to $1-1/e$ on such datasets. From Example 3 and the proofs of Theorems 2 and 3 we can see that the non-submodularity only occurs in the case when two or more cascades arrive at one node at the same time. Thus, if such a scenario does not happen frequently, the Max-$\mathbb{M}$ and Min-$\overline{\mathbb{M}}$ problems will be close to submodular optimization problems, and consequently, the greedy algorithm is effective. While $f(\overline{S}_*)/\overline{f}(\overline{S}_*)$ is data-dependent, we have observed that it is very close to 1 under all the considered datasets, which indicates that the approximation ratio is near-constant.

**Minor observations.** We can also observe that Random offers no help in misinformation containment and HighWeight is also futile in many cases (e.g., Figs. 4a, 5a and 6b where it has the same performance as that of Random). In addition, Proximity performs slightly better than HighWeight does but it can still fail to reduce the number of $\mathbb{M}$-active users when budget increases, i.e., the curve is not monotone decreasing. We have also observed that ALG. 2 strictly outperforms that solely running ALG. 1 on $f_{\overline{\mathbb{M}}}$, which means approximating the upper bound and lower bound can provide better solutions. The results of this part can be found in our supplementary material.

## 7  Conclusion

In this paper, we study the MC problem under the general case where there is an arbitrary number of cascades. The considered scenario is more realistic and it applies to complicated real applications in online social networks. We provide a formal model and address the MC problem from the view of combinatorial optimization. We show the MC problem is not only NP-hard but also admits strong inapproximability property. We propose three types of cascade priority and show that the MC problem can be close to submodular optimization problems. An effective algorithm for solving the MC problem is designed and evaluated by experiments.

## Acknowledgments

This work is supported in part by a start-up grant from the University of Delaware and the NSF under grant #1747818.

## Footnotes

[1]https://www.ndtv.com/mumbai-news/misinformation-on-social-media-led-to-pune-violence-minister-1795562

[2]http://reports.weforum.org/global-risks-2018/digital-wildfires/

[3]When there is only one misinformation cascade and one positive cascade, this problem can be approximated within a factor of $1 - 1/e$ [2, 3, 4]. Informally, the complexity class $DTIME(f(n))$ consists of the decision problems that can be solved in $O(f(n))$.

[4]Note that here $\pi_{t-1}(u)$ cannot be $\emptyset$ so $\mathrm{F}_v(\pi_{t-1}(u))$ is well-defined.

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
