[Supplementary Material]

# On Misinformation Containment in Online Social Networks (Supplementary Material)

**Guangmo (Amo) Tong**
Department of Computer and Information Sciences
University of Delaware
amotong@udel.edu

**Weili Wu**
Department of Computer Science
University of Texas at Dallas
weiliwu@utdallas.edu

**Ding-Zhu Du**
Department of Computer Science
University of Texas at Dallas
dzdu@utdallas.edu

## 1  Proofs

### 1.1  Proofs of Theorems 2 and 3

We first provide some preliminaries. According to the model, with probability $p_{(u,v)}$ that $u$ can successfully activate $v$. We use g to denote a random subgraph sampled from $G$ where each edge $(u,v)$ appears in g with probability $p_{(u,v)}$. Each edge in g then has the propagation probability of 1. We use $\mathcal{G}$ to denote the set of all possible random graphs and use $\Pr[\mathbf{g}]$ to denote the probability that g can be sampled. Let $f^{\mathbf{g}}(\tau(P_*))$ be the number of $\overline{\mathbb{M}}$-active nodes in g under $\tau(P_*)$. Because the randomness of the diffusion process comes from that if each edge $(u,v)$ can be "passed", each graph in $\mathcal{G}$ is actually one possible outcome of the spreading process. Therefore, $f(\tau(P_*))$ can be represented as

$$f(\tau(P_*)) = \sum_{\mathbf{g} \in \mathcal{G}} \Pr[\mathbf{g}] \cdot f^{\mathbf{g}}(\tau(P_*)). \tag{1}$$

Because the properties of monotone nondecreasing and submodular are preserved under addition, to prove Theorem 2 or 3, it suffices to prove that $f^{\mathbf{g}}(\tau(P_*))$ is monotone nondecreasing and submodular.

Let us first consider under which condition a node can be $\overline{\mathbb{M}}$-active or $\mathbb{M}$-active. For each $u,v \in V$ and $V' \subseteq V$, we use $\mathrm{dis}_{\mathbf{g}}(u,v)$ to denote the length of the shortest path from $u$ to $v$ in g, and define that $\mathrm{dis}_{\mathbf{g}}(V',v) = \min_{u \in V'} \mathrm{dis}_{\mathbf{g}}(u,v)$. Let $\tau(\mathbb{M}) = \cup_{M \in \mathbb{M}} \tau(M)$ be the union of the seed sets of misinformation cascades, and define $\tau(\overline{\mathbb{M}}) = (\cup_{P \in \mathbb{P}} \tau(P)) \cup \tau(P_*)$ for the positive cascades. Note that $\tau(\mathbb{M})$ is fixed while $\tau(\overline{\mathbb{M}})$ depends on $\tau(P_*)$. Two important lemmas are given below.

**Lemma 1.** *Under any cascade priority setting, a node $u \in V$ is $\overline{\mathbb{M}}$-active in g if $\mathrm{dis}_{\mathbf{g}}(\tau(\overline{\mathbb{M}}),u) < \mathrm{dis}_{\mathbf{g}}(\tau(\mathbb{M}),u)$.*

*Proof.* Let $a \in \tau(\overline{\mathbb{M}})$ be a node such that $\mathrm{dis}_{\mathbf{g}}(a,u) = \mathrm{dis}_{\mathbf{g}}(\tau(\overline{\mathbb{M}}),u)$, and $(v_0,...,v_l)$ be the shortest path from $a$ to $u$, where $v_0 = a$ and $v_l = u$. Assuming $\mathrm{dis}_{\mathbf{g}}(\tau(\overline{\mathbb{M}}),u) < \mathrm{dis}_{\mathbf{g}}(\tau(\mathbb{M}),u)$, we prove that $v_0,...,v_l$ are all $\overline{\mathbb{M}}$-active and $v_i$ will be activated at time step $i$. We prove this by induction. Because $\mathrm{dis}_{\mathbf{g}}(a,u) = \mathrm{dis}_{\mathbf{g}}(\tau(\overline{\mathbb{M}}),u) < \mathrm{dis}_{\mathbf{g}}(\tau(\mathbb{M}),u)$, $v_0 = a$ is not a seed node of any misinformation cascade, and therefore $v_0$ is $\overline{\mathbb{M}}$-active at time step 0. Suppose that, for some $i$ with $0 < i < l$, $v_0,...,v_i$ are all $\overline{\mathbb{M}}$-active and $v_i$ is activated at time step $i$. Now we prove that $v_{i+1}$ will be $\overline{\mathbb{M}}$-active

at time step $i + 1$. Because $i + 1$ is the length of the shortest path from any seed node to $u$, $u$ cannot be activated before time step $i + 1$. Furthermore, by the inductive hypothesis, $v_i$ will activate $v_{i+1}$ at time step $i + 1$, so $v_{i+1}$ will be activated at time step $i + 1$ by $v_i$ or other in-neighbors. Finally, because $\text{dis}_g(\tau(\overline{\mathbb{M}}), u) < \text{dis}_g(\tau(\mathbb{M}), u)$ and $v_{i+1}$ is on the shortest path from $a$ to $u$, we have $\text{dis}_g(\tau(\overline{\mathbb{M}}), v_{i+1}) < \text{dis}_g(\tau(\mathbb{M}), v_{i+1})$, which means any path from any misinformation seed node to $v_{i+1}$ must have a length larger than $i + 1$. Therefore, $v_{i+1}$ cannot be $M$-active at time step $i + 1$ for any $M \in \mathbb{M}$ and it must be $\overline{\mathbb{M}}$-active. By induction, $u = v_l$ will be $\overline{\mathbb{M}}$-active and it will be activated at time step $l$. $\qquad\square$

We can prove the following lemma in a similar way.

**Lemma 2.** *Under any cascade priority setting, a node $u \in V$ is $\mathbb{M}$-active if $\text{dis}_g(\mathbb{M}, u) < \text{dis}_g(\tau(\overline{\mathbb{M}}), u)$.*

Lemma 1 and 2 give the necessary conditions for a node to be $\mathbb{M}$-active or $\overline{\mathbb{M}}$-active.

### 1.1.1 M-dominant cascade priority

Now let us consider the M-dominant cascade priority. The following lemma shows a necessary and sufficient condition for a node to be $\overline{\mathbb{M}}$-active under the M-dominant cascade priority.

**Lemma 3.** *Under the M-dominant cascade priority, given the seed sets, a node $u$ is $\overline{\mathbb{M}}$-active in $g$ if and only if $\text{dis}_g(\tau(\overline{\mathbb{M}}), u) < \text{dis}_g(\tau(\mathbb{M}), u)$.*

*Proof.* $\Longrightarrow$ : Let $a \in \tau(\mathbb{M})$ be a node such that $\text{dis}_g(a, u) = \text{dis}_g(\tau(\mathbb{M}), u)$, and $(v_0, ..., v_l)$ be the shortest path from $a$ to $u$, where $v_0 = a$ and $v_l = u$. We prove the contrapositive. That is, assuming $\text{dis}_g(\tau(\overline{\mathbb{M}}), u) \geq \text{dis}_g(\tau(\mathbb{M}), u)$, we prove that $v_0, ..., v_l$ are all $\mathbb{M}$-active and $v_i$ will be activated at time step $i$. Again, we prove this by induction. Because $v_0 = a$ is a seed node of some misinformation cascade and the cascade setting is M-dominant, $v_0$ will be $\mathbb{M}$-active at time step $0$. Suppose that, for some $i$ with $0 < i < l$, $v_0, ..., v_i$ are all $\mathbb{M}$-active and $v_i$ is activated at time step $i$. Now we prove that $v_{i+1}$ will be $\mathbb{M}$-active at time step $i + 1$. Because $i + 1$ is the length of the shortest path from any seed node to $v_i$, $v_i$ cannot be activated before time step $i + 1$. Furthermore, by the inductive hypothesis, $v_i$ will activate $v_{i+1}$ at time step $i + 1$ so $v_{i+1}$ will be activated at time step $i + 1$ by $v_i$ or other in-neighbors. Finally, because the cascade priority is M-dominant and $v_i$ is $\mathbb{M}$-active, $v_i$ must be $\mathbb{M}$-active. By induction, $u = v_l$ will be $\mathbb{M}$-active and it will be activated at time step $l$.

$\Longleftarrow$ : This part is exactly the Lemma 1.

$\qquad\square$

Now we are ready to prove that $f^g(\tau(P_*))$ is monotone nondecreasing and submodular. According to Lemma 3, $f^g(\tau(P_*))$ can be expressed as

$$f^g(\tau(P_*)) = \sum_{u \in V} f^g(\tau(P_*), u),$$

where $f^g(\tau(P_*), u)$ is defined as

$$f^g(\tau(P_*), u) = \begin{cases} 1 & \text{if } \text{dis}_g(\tau(\overline{\mathbb{M}}), u) < \text{dis}_g(\tau(\mathbb{M}), u) \\ 0 & \text{else} \end{cases}, \tag{2}$$

where $\tau(\overline{\mathbb{M}})$ depends on $\tau(P_*)$. Now it suffices to prove that $f^g(\tau(P_*), u)$ is monotone nondecreasing and submodular with respect to $\tau(P_*)$.

**Lemma 4.** *$f^g(\tau(P_*), u)$ is monotone nondecreasing and submodular for each $g \in \mathcal{G}$ and $u \in V$.*

*Proof.* It is clear monotone nondecreasing because adding one node to $\tau(P_*)$ will not increase $\text{dis}_g(\tau(\overline{\mathbb{M}}), u)$. To prove the submodularity, it suffices to prove that for each $S_1 \subseteq S_2 \subseteq V^*$ and $x \notin S_2$,

$$f^g(S_1 \cup \{x\}, u) - f^g(S_1, u) \geq f^g(S_2 \cup \{x\}, u) - f^g(S_2, u).$$

Because $f^{\mathrm{g}}(\tau(P_*), u)$ is monotone nondecreasing and it can be only 0 or 1, it suffices to show that $f^{\mathrm{g}}(S_1 \cup \{x\}, u) - f^{\mathrm{g}}(S_1, u) = 1$ whenever $f^{\mathrm{g}}(S_2 \cup \{x\}, u) - f^{\mathrm{g}}(S_2, u) = 1$. If $f^{\mathrm{g}}(S_2 \cup \{x\}, u) - f^{\mathrm{g}}(S_2, u) = 1$, then $f^{\mathrm{g}}(S_2 \cup \{x\}, u) = 1$ and $f^{\mathrm{g}}(S_2, u) = 0$. Because $f^{\mathrm{g}}(S_2, u) = 0$ and $\mathrm{dis}_{\mathrm{g}}(S_1, u) \geq \mathrm{dis}_{\mathrm{g}}(S_2, u)$, we have $f^{\mathrm{g}}(S_1, u) = 0$ . Because $f^{\mathrm{g}}(S_2 \cup \{x\}, u) = 1$ and $f^{\mathrm{g}}(S_2, u) = 0$, by Eq. (2), we have $\mathrm{dis}_{\mathrm{g}}(x, u) < \mathrm{dis}_{\mathrm{g}}(\tau(\mathbb{M}), u)$. Therefore, $\mathrm{dis}_{\mathrm{g}}(S_1 \cup \{x\}, u) < \mathrm{dis}_{\mathrm{g}}(\tau(\mathbb{M}), u)$ and $f^{\mathrm{g}}(S_1 \cup \{x\}, u) = 1$. So $f^{\mathrm{g}}(S_1 \cup \{x\}, u) - f^{\mathrm{g}}(S_1, u)$ is also equal to 1.

$\square$

### 1.1.2 P-dominant cascade priority

Now we prove the P-dominant case. The proof is similar to that of the M-dominant case. We use the following lemma analogous to Lemma 3.

**Lemma 5.** *Under the P-dominant cascade priority, a node $u$ is $\overline{\mathbb{M}}$-active in $\mathrm{g}$ if and only if* $\mathrm{dis}_{\mathrm{g}}(\tau(\overline{\mathbb{M}}), u) \leq \mathrm{dis}_{\mathrm{g}}(\tau(\mathbb{M}), u)$.

*Proof.* $\impliedby$ : Let $a \in \tau(\overline{\mathbb{M}})$ be the node such that $\mathrm{dis}_{\mathrm{g}}(a, u) = \mathrm{dis}_{\mathrm{g}}(\tau(\overline{\mathbb{M}}), u)$, and $(v_0, ..., v_l)$ be the shortest path from $a$ to $u$ where $v_0 = a$ and $v_l = u$. Assuming $\mathrm{dis}_{\mathrm{g}}(\tau(\overline{\mathbb{M}}), u) \leq \mathrm{dis}_{\mathrm{g}}(\tau(\mathbb{M}), u)$, we can prove that $v_0, ..., v_l$ are all $\overline{\mathbb{M}}$-active and $v_i$ will be activated at time step $i$. This is similar to the " $\implies$ " part in the proof of Lemma 3.

$\implies$ : This part follows from Lemma 2. $\square$

Therefore, for the P-dominant case, $f^{\mathrm{g}}(\tau(P_*))$ can be represented as $f^{\mathrm{g}}(\tau(P_*)) = \sum_{u \in V} \cdot f^{\mathrm{g}}(\tau(P_*), u)$, where $f^{\mathrm{g}}(\tau(P_*), u)$ is defined as

$$f^{\mathrm{g}}(\tau(P_*), u) = \begin{cases} 1 & \text{if } \mathrm{dis}_{\mathrm{g}}(\tau(\mathbb{M}), u) \leq \mathrm{dis}_{\mathrm{g}}(\tau(\overline{\mathbb{M}}), u) \\ 0 & \text{else} \end{cases}. \tag{3}$$

Now it suffices to prove that $f^{\mathrm{g}}(\tau(P_*), u)$ is monotone nondecreasing and submodular.

**Lemma 6.** $f^{\mathrm{g}}(\tau(P_*), u)$ *is monotone nondecreasing and submodular for each $\mathrm{g} \in \mathcal{G}$ and $u \in V$.*

*Proof.* It is clear monotone nondecreasing as adding one node to $\tau(P_*)$ will not increase $\mathrm{dis}_{\mathrm{g}}(\tau(\mathbb{M}), u)$. To prove submodularity, it suffices to prove that for each $S_1 \subseteq S_2 \subseteq V^*$ and $x \notin S_2$,

$$f^{\mathrm{g}}(S_1 \cup \{x\}, u) - f^{\mathrm{g}}(S_1, u) \geq f^{\mathrm{g}}(S_2 \cup \{x\}, u) - f^{\mathrm{g}}(S_2, u).$$

It suffices to show that $f^{\mathrm{g}}(S_1 \cup \{x\}, u) - f^{\mathrm{g}}(S_1, u) = 1$ whenever $f^{\mathrm{g}}(S_2 \cup \{x\}, u) - f^{\mathrm{g}}(S_2, u) = 1$. If $f^{\mathrm{g}}(S_2 \cup \{x\}, u) - f^{\mathrm{g}}(S_2, u) = 1$, then $f^{\mathrm{g}}(S_2 \cup \{x\}, u) = 1$ and $f^{\mathrm{g}}(S_2, u) = 0$. Because $f^{\mathrm{g}}(S_2, u) = 0$ and $\mathrm{dis}_{\mathrm{g}}(S_1, u) \geq \mathrm{dis}_{\mathrm{g}}(S_2, u)$, we have $f^{\mathrm{g}}(S_1, u) = 0$. Because $f^{\mathrm{g}}(S_2 \cup \{x\}, u) = 1$ and $f^{\mathrm{g}}(S_2, u) = 0$, by Eq. (3), $\mathrm{dis}_{\mathrm{g}}(x, u) \leq \mathrm{dis}_{\mathrm{g}}(\tau(\mathbb{M}), u)$. Therefore, $\mathrm{dis}_{\mathrm{g}}(S_1 \cup \{x\}, u) \leq \mathrm{dis}_{\mathrm{g}}(\tau(\mathbb{M}), u)$ and $f^{\mathrm{g}}(S_1 \cup \{x\}, u) = 1$. So $f^{\mathrm{g}}(S_1 \cup \{x\}, u) - f^{\mathrm{g}}(S_1, u)$ is also equal to 1. $\square$

### 1.1.3 Homogeneous cascade priority

Since we are considering the homogeneous cascade priority, we denote $\mathrm{F}_v()$ as $\mathrm{F}()$ without mentioning any node. For each $u \in V$, $\mathrm{g} \in \mathcal{G}$ and $\tau(P_*) \subseteq V^*$, let

$$A_{\mathrm{g}}(\tau(P_*), u) = \{v \in \tau(\mathbb{M}) \cup \tau(\overline{\mathbb{M}}) | \mathrm{dis}_{\mathrm{g}}(v, u) = \mathrm{dis}_{\mathrm{g}}(\tau(\mathbb{M}) \cup \tau(\overline{\mathbb{M}}), u)\}$$

be the set of the node $v$ such that $\mathrm{dis}_{\mathrm{g}}(v, u) = \mathrm{dis}_{\mathrm{g}}(\tau(\mathbb{M}) \cup \tau(\overline{\mathbb{M}}), u)$. Let

$$\mathbb{C}_{\mathrm{g}}(\tau(P_*), u) = \{C | C \in \mathbb{M} \cup \mathbb{P} \cup \{P_*\}, \tau(C) \cap A_{\mathrm{g}}(\tau(P_*), u) \neq \emptyset\}$$

be set of the cascade(s) with a seed node in $A_{\mathrm{g}}(\tau(P_*), u)$. Let $C_{\mathrm{g}}(\tau(P_*), u) \in \mathbb{C}_{\mathrm{g}}(\tau(P_*), u)$ be the cascade such that

$$C_{\mathrm{g}}(\tau(P_*), u) = \underset{C \in \mathbb{C}_{\mathrm{g}}(\tau(P_*), u)}{\arg\max} \ \mathrm{F}(C).$$

**Lemma 7.** *Under the homogeneous cascade priority, each node $u \in V$ will be $C_{\mathrm{g}}(\tau(P_*), u)$-active in $\mathrm{g}$ under $\tau(P_*)$.*

*Proof.* Let $a$ be an arbitrary node in $\tau(C_{\mathrm{g}}(\tau(P_*), u)) \cap A_{\mathrm{g}}(\tau(P_*), u)$,[1] and $(v_0 = a, ..., v_l = u)$ be the shortest path from $a$ to $u$ where $v_0 = a$ and $v_l = u$. We prove that $v_0, ..., v_l$ are all $C_{\mathrm{g}}(\tau(P_*), u)$-active and $v_i$ will be activated at time step $i$. We prove this by induction.

**Basic step:** Because $a$ belongs to $A_{\mathrm{g}}(\tau(P_*), u)$, any cascade selecting $a$ as a seed node must in $\mathbb{C}_{\mathrm{g}}(\tau(P_*), u)$. Since $C_{\mathrm{g}}(\tau(P_*), u)$ has the highest priority among $\mathbb{C}_{\mathrm{g}}(\tau(P_*), u)$, $a = v_0$ will be $C_{\mathrm{g}}(\tau(P_*), u)$-active at time step 0.

**Inductive step:** Suppose that, for some $0 < i < l$, $v_0, ..., v_i$ are all $C_{\mathrm{g}}(\tau(P_*), u)$-active and $v_i$ is activated at time step $i$. Now we prove that $v_{i+1}$ will be $C_{\mathrm{g}}(\tau(P_*), u)$-active at time step $i+1$. Because $i+1$ is the length of the shortest path from any seed node to $v_{i+1}$, $v_{i+1}$ cannot be activated before time step $i+1$. Furthermore, $v_i$ will activate $v_{i+1}$ at time step $i+1$, so $v_{i+1}$ will be activated at time step $i+1$ by $v_i$ or other in-neighbors. Note that only the cascades with seed nodes in $A_{\mathrm{g}}(\tau(P_*), u)$ are able to activate $v_{i+1}$ at time step $i+1$. Because $v_i$ is $C_{\mathrm{g}}(\tau(P_*), u)$-active and $C_{\mathrm{g}}(\tau(P_*), u)$ has the highest priority among $A_{\mathrm{g}}(\tau(P_*), u)$, $v_{i+1}$ will be activated by $v_i$ at time step $i+1$ and it will be $C_{\mathrm{g}}(\tau(P_*), u)$-active. $\qquad\square$

According to Lemma 7, a node $u$ is $\overline{\mathbb{M}}$-active in g under $\tau(P_*)$ if and only if $C_{\mathrm{g}}(\tau(P_*), u) \in \mathbb{P} \cup \{P_*\}$. Therefore, under the homogeneous cascade setting, $f^{\mathrm{g}}(\tau(P_*))$ can be represented as $f^{\mathrm{g}}(\tau(P_*)) = \sum_{u \in V} f^{\mathrm{g}}(\tau(P_*), u)$, where $f^{\mathrm{g}}(\tau(P_*), u)$ is defined as

$$f^{\mathrm{g}}(\tau(P_*), u) = \begin{cases} 1 & \text{if } C_{\mathrm{g}}(\tau(P_*), u) \in \mathbb{P} \cup \{P_*\} \\ 0 & \text{else, } C_{\mathrm{g}}(\tau(P_*), u) \in \mathbb{M} \end{cases}.$$

Now it suffices to prove that $f^{\mathrm{g}}(S, u)$ is monotone nondecreasing and submodular.

**Lemma 8.** $f^{\mathrm{g}}(\tau(P_*), u)$ *is monotone nondecreasing for each* $\mathrm{g} \in \mathcal{G}$ *and* $u \in V$.

*Proof.* When a new node $x$ is added to $\tau(P_*)$, $\mathrm{dis}_{\mathrm{g}}(\tau(C), u)$ remains unchanged for $C \neq P_*$, and $\mathrm{dis}_{\mathrm{g}}(\tau(P_*), u)$ either decreases or remains unchanged. Therefore, after adding a new node to $\tau(P_*)$, there are three possible cases. First, $P_*$ becomes a new cascade in $\mathbb{C}_{\mathrm{g}}(\tau(P_*), u)$. Second, $P_*$ is the only cascade in $\mathbb{C}_{\mathrm{g}}(\tau(P_*), u)$. Third, $\mathbb{C}_{\mathrm{g}}(\tau(P_*), u)$ remains unchanged. In either of the three cases, $C_{\mathrm{g}}(\tau(P_*), u)$ cannot change to a misinformation cascade from a positive cascade. Therefore, $f^{\mathrm{g}}(\tau(P_*), u)$ is monotone nondecreasing. $\qquad\square$

**Lemma 9.** $f^{\mathrm{g}}(\tau(P_*), u)$ *is submodular for each* $\mathrm{g} \in \mathcal{G}$ *and* $u \in V$.

*Proof.* Again, it suffices to prove that for each g, $S_1 \subseteq S_2 \subseteq V^*$ and $x \notin S_2$,

$$f^{\mathrm{g}}(S_1 \cup \{x\}, u) - f^{\mathrm{g}}((S_1, u)) \geq f^{\mathrm{g}}(S_2 \cup \{x\}, u) - f^{\mathrm{g}}(S_2, u).$$

Furthermore, it suffices to show that $f^{\mathrm{g}}(S_1 \cup \{x\}, u) - f^{\mathrm{g}}(S_1, u) = 1$ whenever $f^{\mathrm{g}}(S_2 \cup \{x\}, u) - f^{\mathrm{g}}(S_2, u) = 1$. If $f^{\mathrm{g}}(S_2 \cup \{x\}, u) - f^{\mathrm{g}}(S_2, u) = 1$, then $f^{\mathrm{g}}(S_2 \cup \{x\}, u) = 1$ and $f^{\mathrm{g}}(S_2, u) = 0$. Because $f^{\mathrm{g}}(S_2, u) = 0$, we have $C_{\mathrm{g}}(S_2, u) \in \mathbb{M}$. Since $S_1 \subseteq S_2$, by the discussion in the proof of Lemma 8, $C_{\mathrm{g}}(S_1, u)$ also belongs $\mathbb{M}$ and therefore $f^{\mathrm{g}}(S_1, u) = 0$. Because $f^{\mathrm{g}}(S_2 \cup \{x\}, u) = 1$ and $f^{\mathrm{g}}(S_2, u) = 0$, we have $C_{\mathrm{g}}(S_2, u) \in \mathbb{M}$ and $C_{\mathrm{g}}(S_2 \cup \{x\}, u) \in \mathbb{P}$. According to the three cases in the proof of Lemma 8, $C_{\mathrm{g}}(S_2 \cup \{x\}, u)$ must be $P_*$. Therefore, $C_{\mathrm{g}}(S_1 \cup \{x\}, u)$ is also $P_*$ and $f^{\mathrm{g}}(S_1 \cup \{x\}, u) = 1$. So $f^{\mathrm{g}}(S_1 \cup \{x\}, u) - f^{\mathrm{g}}(S_1, u)$ is also equal to 1. $\qquad\square$

## 1.2 Proof of Theorem 4

The proof of Theorem 4 requires the preliminaries given in the last subsection. For each $\mathrm{g} \in \mathcal{G}$ and $u \in V$, let us define

$$f^{\mathrm{g}}(\tau(P_*), u) = \begin{cases} 1 & \text{if } u \text{ is } \overline{\mathbb{M}}\text{-active under } \tau(P_*) \text{ in g with cascade priority } \mathrm{F}_v \text{ for each } v \\ 0 & \text{else} \end{cases},$$

$$\overline{f}^{\mathrm{g}}(\tau(P_*), u) = \begin{cases} 1 & \text{if } u \text{ is } \overline{\mathbb{M}}\text{-active under } \tau(P_*) \text{ in g with cascade priority } \overline{\mathrm{F}}_v \text{ for each } v \\ 0 & \text{else} \end{cases},$$

and

$$\underline{f}^{\mathrm{g}}(\tau(P_*), u) = \begin{cases} 1 & \text{if } u \text{ is } \overline{\mathbb{M}}\text{-active under } \tau(P_*) \text{ in } \mathrm{g} \text{ with cascade priority } \underline{F}_v \text{ for each } v \\ 0 & \text{else} \end{cases}.$$

According to Eq. (1), it suffices to show that $\overline{f}^{\mathrm{g}}(\tau(P_*), u) \geq f^{\mathrm{g}}(\tau(P_*), u) \geq \underline{f}^{\mathrm{g}}(\tau(P_*), u)$.

When $\underline{f}^{\mathrm{g}}(\tau(P_*), u) = 1$, by Lemma 3, we have $\mathrm{dis}_{\mathrm{g}}(\tau(\overline{\mathbb{M}}), u) < \mathrm{dis}_{\mathrm{g}}(\tau(\mathbb{M}), u)$. According to Lemma 1, $u$ will be $\overline{\mathbb{M}}$-active under $F_v$ and thus $f^{\mathrm{g}}(\tau(P_*), u)$ must be 1. Therefore, $f^{\mathrm{g}}(\tau(P_*), u) \geq \underline{f}^{\mathrm{g}}(\tau(P_*), u)$.

When $\overline{f}^{\mathrm{g}}(\tau(P_*), u) = 0$, by Lemma 5, we have $\mathrm{dis}_{\mathrm{g}}(\tau(\overline{\mathbb{M}}), u) > \mathrm{dis}_{\mathrm{g}}(\tau(\mathbb{M}), u)$. According to Lemma 2, $u$ will be $\mathbb{M}$-active under $F_v$ and thus $f^{\mathrm{g}}(\tau(P_*), u)$ must be 0. Therefore, $\overline{f}^{\mathrm{g}}(\tau(P_*), u) \geq f^{\mathrm{g}}(\tau(P_*), u)$.

## 2 Experiments

The statistics of the datasets are listed in Table 1

Table 1: Datasets

| Dataset | Node | Edge |
|---------|------|------|
| Higgs-10K | 10,000 | 22,482 |
| Higgs-100K | 100,000 | 193,484 |
| HepPh | 34,000 | 421,578 |

### 2.1 Experimental setting

For the case of five cascades, we deploy two existing misinformation cascades and two existing positive cascades. For the case of ten cascades, we deploy four existing misinformation cascades and five existing positive cascades. For each existing cascade, the size of the seed set is set as 20 and the seed nodes are selected from the node with the highest single-node influence. The seed sets of different cascades do not overlap with each other. The budget of $P_*$ is enumerated from $\{1, 2, ..., 20\}$. The cascade priority at each node is assigned randomly, by generating random permutations. The lists of the used random permutations over $\{1, 2, 3\}$, $\{1, ..., 5\}$ and $\{1, ..., 10\}$ are provided in the supplementary material. For each dataset, the list of the nodes ordered by single-node influence is provided in the supplementary material.

### 2.2 More experimental results

The performance of ALG. 1 on $\overline{f}$, $f$ and $\underline{f}$ on each dataset is shown in Figs. 1, 2 and 3. As we can see in the figures, maximizing the upper or lower bound may provide a better solution that just maximizing the objective function. Therefore, the sandwich algorithm is effective than the naive greedy algorithm on $f$.

(a) Three cascades      (b) Five cascades      (c) Ten cascades

Figure 1: Results on Higgs-10K.

(a) Three cascades     (b) Five cascades     (c) Ten cascades

Figure 2: Results on Higgs-100K.

(a) Three cascades     (b) Five cascades     (c) Ten cascades

Figure 3: Results on HepPh.

## Footnotes

[1] By definition, $\tau(C_{\mathrm{g}}(\tau(P_*), u)) \cap A_{\mathrm{g}}(\tau(P_*), u)$ cannot be empty.