[Reviews · NeurIPS 2018]

Reviewer 1



Summary The authors study the problem of cascade containment. The main idea is that if we have "bad" cascades, we can start "good" cascades to "inoculate" individuals against the bad ones. This results here (1) the problem is worst-case hard, 2) approximate algorithms exist) are related to classical results in influence maximization but differ in that now we aren't just trying to maximize the size of a cascade. Review I'm not really sure how to review this paper. I am not an expert in the IM/graph cascade area, so I cannot judge the novelty/correctness of the theorems/algorithms presented here. However, as someone who has studied this phenomenon empirically, I'm unconvinced that the mathematical model corresponds to the real-world problem of misinformation cascades in social networks. 1) I'm not sure the "innoculation" model is actually a model of how people process misinformation. That is, just revealing that a particular fact (that happens to be aligned with my current beliefs) isn't true is unlikely to make me not believe the fact. 2) I'm not sure I understand who this mechanism is designed for. Is the idea that social networks should find the right people to seed to create these positive cascades? Is that a realistic policy for social networks to actually start doing?

Reviewer 2



This paper addresses the problem of minimizing the effect of negative (e.g., false, harmful) information cascades in online social networks, i.e. misinformation containment. In particular, they consider the case of an arbitrary number of cascades, whereas most prior work in NIPS, KDD, INFOCOM, etc. have focused on the cases of one or two cascades. The authors provide a formal model for multi-cascade influence diffusion and cascade priority (Section 3), design an algorithm for minimizing the number of affected nodes (Section 5), provide theoretical bounds in terms of runtime and optimality (Section 4), and demonstrate the superiority of their algorithm against baselines on real-world datasets. Overall, I think this is a very good submission. The authors provide clear motivation for the multi-cascade misinformation problem, rigorous theory around the proposed algorithm, and a reasonably comprehensive set of experiments on three real-world datasets. The paper is both original and significant because at least to my knowledge, the multi-cascade problem in online social networks has not been mathematically modeled before. Further, the paper is well written, with all ideas and even theorems explained with clarity. For these reasons, I recommend accepting the paper. At the same time, I also found two areas for improvement. First, I would encourage the authors to consider a more comprehensive set of propagation probabilities in the experimentation. As it stands, it seems that these are set deterministically based on the observed social network, e.g., the number of activities observed from u to v and the out degree of particular nodes. This is not likely to capture nuances in how people influence one another, though, including the susceptibility of individuals, the influential power of individuals, and the fraction of interactions that are actually attempting to spread information. There is of course no way to account for all of these factors, but one idea would be to add random Gaussian noise on each of the link weights, and average the results over multiple trials, to see how robust the algorithms are to fluctuations. Another possibility would be to weight each of the probabilities by a measure of importance of the root node, like their social centrality. Second, I feel that the authors refer to the supplemental material too often. To rectify this, I would suggest moving Section 4 and a few of the less important theorems entirely to the supplemental material, and instead including more of the proofs from Sections 3 and 5 in the manuscript itself. This way the paper would be more self-contained, and the reader does not have to flip back and forth between the paper and the supplement as much.

Reviewer 3



The paper deals with the misinformation spread in online social networks where multiple competing "positive" and "negative" (misinformation) campaigns are launched simultaneously and arbitrary number of cascades is allowed. The misinformation spread then is modeled as a combinatorial optimization where the number of "negative" cascades is minimized. Strengths - The novelty of the paper is the extension of existing work on modeling misinformation propagation in social media to multiple cascades run simultaneously. - The problem of the misinformation propagation in social media is significant and timely given the recent proved intervention of bots and social influencers in national elections. - The model of cascade priority is interesting and clearly defined. Weekenesses - The paper lucks clarity and some bits need to be explained more rigorously. For example, in the abstract DTIME is introduced without a definition of what is this. Similarly, in the introduction the concepts of M and P-dominant cascade priorities are mentioned while they are only explained in later sections. - Evaluation is pretty limited. It could be extended to cover a broader range of news stories (and topics) on Twitter. For example, claims made by candidates during the US Presidential election, Brexit or the global warming etc.. You could also focus on evaluating your model on the misinformation diffusion for different "fake news" or propaganda outlets and compare the differences. Other Comments - Line 21: Reference to the World Economic Forum global risks is missing - Line 37: What do you mean by different sources? User accounts or media sources? - Line 60: "nontrivial upper bound". What a trivial upper bound would be? - Lines 74-75: Sentence needs to be rephrased - Lines 99-101: You could mention that cascade priority will be defined later in section 5.1